# Occupational Radiation Dose to Eye Lenses in CT-Guided Interventions Using MDCT-Fluoroscopy

**DOI:** 10.3390/diagnostics11040646

**Published:** 2021-04-02

**Authors:** Yohei Inaba, Shin Hitachi, Munenori Watanuki, Koichi Chida

**Affiliations:** 1Course of Radiological Technology, Health Sciences, Tohoku University Graduate School of Medicine, 2-1 Seiryo, Aoba, Sendai, Miyagi 980-8575, Japan; chida@med.tohoku.ac.jp; 2Department of Radiation Disaster Medicine, International Research Institute of Disaster Science, Tohoku University, 468-1 Aramaki Aza-Aoba, Aoba, Sendai, Miyagi 980-0845, Japan; 3Department of Radiology, Tohoku University Hospital, 1-1 Seiryo, Aoba, Sendai, Miyagi 980-8575, Japan; hitachi@rad.med.tohoku.ac.jp; 4Department of Orthopaedic Surgery, Tohoku University Hospital, 1-1 Seiryo, Aoba, Sendai, Miyagi 980-8575, Japan; munenori.watanuki.e2@tohoku.ac.jp

**Keywords:** CT-guided interventions, eye lens, occupational radiation dose, radiation protection, scattered radiation

## Abstract

In computed tomography (CT)-guided interventions (CTIs), physicians are close to a source of scattered radiation. The physician and staff are at high risk of radiation-induced injury (cataracts). Thus, dose-reducing measures for physicians are important. However, few previous reports have examined radiation doses to physicians in CTIs. This study evaluated the radiation dose to the physician and medical staff using multi detector (MD)CT-fluoroscopy, and attempted to understand radiation-protection and -reduction methods. The procedures were performed using an interventional radiology (IVR)-CT system. We measured the occupational radiation dose (physician and nurse) using a personal dosimeter in real-time, gathered CT-related parameters (fluoroscopy time, mAs, CT dose index (CTDI), and dose length product (DLP)), and performed consecutive 232 procedures in CT-guided biopsy. Physician doses (eye lens, neck, and hand; μSv, average ± SD) in our CTIs were 39.1 ± 36.3, 23.1 ± 23.7, and 28.6 ± 31.0, respectively. Nurse doses (neck and chest) were lower (2.3 ± 5.0 and 2.4 ± 4.4, respectively) than the physician doses. There were significant correlations between the physician doses (eye and neck) and related factors, such as CT-fluoroscopy mAs (eye dose: r = 0.90 and neck dose: r = 0.83). We need to understand the importance of reducing/optimizing the dose to the physician and medical staff in CTIs. Our study suggests that physician and staff doses were not significant when the procedures were performed with the appropriate radiation protection and low-dose techniques.

## 1. Introduction

The clinical benefits of computed tomography (CT)-guided interventions (CTIs) are clear, and these procedures are increasing in number. In CTIs, procedure instruments, such as biopsy needles, are used under CT-fluoroscopic control, including CT acquisition. Using multi detector (MD)-CT in CTIs is useful because multiple CT fluoroscopic images are obtained during the placement of a needle. Thus, many studies have shown that CTIs are a safe and effective guidance tool for percutaneous interventional procedures compared to those using ultrasonography (US) [1,2,3,4,5]. CT-fluoroscopy has the potential to improve the efficacy of CTIs, to reduce the time to place a needle into a lesion for the performance of a biopsy, and to reduce procedure times [6,7,8,9,10].

On the other hand, during CTI procedures, physicians are close to the source of scattered radiation. One of the concerns with the use of CT-fluoroscopy is the high radiation exposure. The physician and medical staff are at high risk of tissue reaction (such as cataracts) at locations close to the scanning plane. Thus, dose-reducing measures for physicians are important [11,12,13]. In 2012, the International Commission on Radiological Protection (ICRP) recommended that the occupational dose limit to the eye lens be reduced by approximately 1/8 (from 150 mSv/year to 20 mSv/year, averaged over a defined 5 years, with not exceeding 50 mSv in any year) [14]. Moreover, the International Atomic Energy Agency (IAEA) has adopted the new eye dose limit by ICRP [15], and many countries are considering adoption this regulation. In Japan, since the new eye dose limit will be adopted on April 2021, the evaluation of occupational dose is important. Several studies have investigated the occupational dose to eye lens undergoing X-ray examinations, such as interventional radiology (IVR) [16,17,18,19,20,21,22,23,24]. However, few previous reports have examined radiation doses to physicians and nurses during CTIs in real-time. The purposes of this study were to measure and analyze the occupational radiation dose to the physicians and nurses during CTIs using MDCT-fluoroscopy, and to understand the importance of radiation-protection and -reduction methods for physicians in CTIs.

## 2. Materials and Methods

The procedures were performed using an IVR-CT system (Equipped MDCT-Aquilion LB; Toshiba, Otawara, Japan). This study was conducted at Tohoku University Hospital (Sendai, Japan). For measurement of scattered radiation distribution, we used a 1-cm dose equivalent i2 dosimeter (RaySafe i2; Unfors RaySafeTM, Billdal, Sweden) in real-time. The sensitivity of the i2 dosimeter ranged from 40 μSv/h to 300 mSv/h.

### 2.1. Scattered Radiation Distribution in the Phantom Study

We measured the planar and height distribution of scattered radiation using an anthropomorphic phantom (PBU-60; KYOTO KAGAKU, Kyoto, Japan) on the patient couch at off-center 90 cm. Measurement points of planar distribution were 19 points at intervals of 60 cm from the gantry center (Figure 1). In addition, the geometric arrangement of radiation height distribution had 3 height points (100, 150, 200 cm) from the floor. The X-ray conditions of CT-fluoroscopy used were 120 kV-p tube voltage, 20 mA tube current, 0.5 s rotation time, 4 mm slice thickness ×3 cross-sections, and 10 s fluoroscopic time.

In order to determine the protective effect of lead acrylic shields suspended from the ceiling (KYOWAGLAS-XA, lead equivalent: 0.5 mmPb, 65 × 70 cm^2^; Osaka, Japan), we measured scattered radiation at various heights (10, 20, 30, 40, 50, and 70 cm), and i2 dosimeter was placed at position ① of 150 cm height from the floor (Figure 2). The shielding ratio was evaluated by the following Equation (1).
(1)$$\mathrm{Shielding}\;\mathrm{ratio}\;\left[\%\right]\;\;=\;\left(1\;-\;\;\frac{\mathrm{Shielding}\;\mathrm{value}}{\mathrm{No}\;\mathrm{shielding}\;\mathrm{value}}\right)\;\times \;100$$

### 2.2. Occupational Doses (Physician and Nurse) in a Clinical Setting

We evaluated the occupational radiation doses (physician and nurse) and CT-related parameters (fluoroscopic time, fluoroscopic mAs, CT dose index (CTDI) and dose length product (DLP)) for 232 consecutive procedures in CT-guided biopsy performed from May 2014 to January 2017. For measurement of occupational dose per procedure, we used a personal i2 dosimeter (RaySafe i2) in real-time, and a pocket dosimeter (PDM127, Hitachi Aloka Medical, Tokyo, Japan). The personal dosimeters were small (4.5 × 4.5 × 1.0 cm^3^), which makes them easy to carry during interventions. The base station displays the dose rates and cumulative dose (monitoring maximum of eight). The physician dose during CTIs was monitored for the eyes, neck, and hands using six i2 dosimeters in real-time, and at the inside and outside of the lead apron at the chest to estimate the effective dose using two pocket dosimeters (Figure 3). The physicians did not use protective eyeglasses and lead acrylic shields in this clinical setting. The nurse doses were also monitored for the neck and chest using two i2 dosimeters in real-time. The effective dose was evaluated by the following Equation (2), Ha; outside dose of lead apron, Hb; inside dose of lead apron.
(2)Effective dose  =  0.11Ha + 0.89Hb

### 2.3. Statical Analysis

Pearson correlation coefficient analysis was used to determine whether CT-related parameters were linearly related to occupational doses Weltch’s t-test was used to compare between dosage groups for bone and soft tissue and to compare between dosage groups without and with a lead drape. 

All statistical analyses were performed using JMP Pro 15 software (SAS Institute Inc., Cary, NC, USA). The statistical significance was defined as *p* < 0.05.

## 3. Results

### 3.1. Scattered Radiation Distribution in the Phantom Study

This study clarified the scattered radiation for planar and height distribution using a phantom (Figure 4). Red circles are the high scattered radiation at approximately 3000 μSv/h near the gantry. The gantry sides scattered radiation was 0 μSv/h. Figure 4b indicates the section of point 13, 1, 2 and 3 at a height of 100, 150, and 200 cm. The amount of scattered radiation in the height direction showed almost the same value.

The shielding ratio regarding height changes was approximately 80% until a 40 cm height, but at a 50 cm height was very low (Figure 5). Therefore, the lead acrylic shields suspended from the ceiling need to be set up at an appropriate position at eye lens level.

### 3.2. Occupational Doses (Physician and Nurse) in a Clinical Setting

Table 1 shows the details of locations and eye dose at each location for 232 procedures in CT-guided biopsies.

Acquisitions numbers, fluoroscopic time, fluoroscopic mAs (average ± SD) of the CTIs were 34.7 ± 23.1, 26.6 ± 17.8 s, and 650.0 ± 598.4, respectively. Physician doses (eye lens, neck, and hands; μSv, average ± SD) in our CTIs were 39.1 ± 36.3, 23.1 ± 23.7, and 28.6 ± 31.0, respectively. Nurse doses (neck and chest) were lower (2.3 ± 5.0 and 2.4 ± 4.4, respectively) than the physician doses (Table 2). 

The correlation between the occupational dose and CT dose-related factors revealed significant correlations between the physician doses (eye lens, neck, and hands) and dose-related factors, such as CT-fluoroscopic mAs (eye lens dose: r = 0.90, neck dose: r = 0.83, and hand dose: r = 0.75). By contrast, there was no significant correlation between occupational doses and CTDI vol, DLP. There were no significant correlations between nurse doses and related factors (Table 3).

A comparison of the radiation doses and CT dose-related factors in bone and soft tissue biopsies is shown in Table 4. In MDCT-guided interventions, physician dose and related factors for bone biopsies tended to be higher than for soft tissue biopsies, although they were no significant differences. A comparison of the radiation doses and CT dose-related factors between without and with lead drape in CTIs is shown in Table 5. Physician doses (hand; *p* = 0.0002 and effective dose; *p* = 0.0008) without a lead drape were absolutely higher than with a drape, in spite of no significant differences in CT dose-related factors. Therefore, it is better to place it over the patients to reduce the physician’s hand dose when possible. There were no significant differences between without and with a lead drape for nurse dose.

## 4. Discussion

It has been reported that the monitoring and protection of occupational dose to the eye lenses in various X-ray examinations is important [25,26,27,28,29,30,31,32,33,34]. In this study, the scattered radiations near the gantry were approximately 3000 μSv/h. The eye lens dose for physicians could reach the revised eye equivalent dose limit (20 mSv/y) in about 90 min if the appropriate protection tools were not used. Moreover, since the amount of scattered radiation in the height direction was almost the same value, attention should be given to the radiation exposure of physicians at all height points. Therefore, the physician needs to optimize the eye lens protection, such as using lead acrylic shields or lead glass eye shields to reduce the lens exposure. We also recommend that medical staff such as nurses move to the gantry sides (0 mSv/h) as much as possible for reducing exposure to radiation. In Figure 5, although the shielding ratio was high until 40 cm height, at a 50 cm height it was very low. Thus, a set up best suited for lead acrylic shields at an eye lens level needs to be implemented when possible.

From our clinical results, we revealed that the maximum eye lens dose for a physician was about 300 μSv/procedure if the physician is close to the source of scattered radiation without radiation protective tools. Therefore, approximately 70 procedures/year have the potential to exceed the eye lens limit (20 mGy/year). However, when assessed by average eye dose (about 40 μSv/procedure), the new eye dose limit would be reached in approximately 500 procedures/year. Some studies have reported that the shielding effect of the radiation protective eyeglasses was approximately 50~60% [16,22,23]. In this way, if protective eyeglasses are used during CTIs, up to about 1000 procedures could be performed annually. Thus, the physician doses were not significant when the procedures were performed with appropriate radiation protection. In addition, in Table 3, there were significant correlations between physician doses and CT dose-related factors, especially CT-fluoroscopic mAs. We recommend minimizing the number of acquisitions and using the lower-dose (mA) mode in order to reduce the physician dose. By contrast, the nurse dose had no significant correlation with CT dose-related factors. Since the nurse dose cannot be estimated from CT dose-related factors, the nurse should be in an appropriate position such as the CT gantry sides to reduce exposure to scattered radiation.

A comparison between other studies and our study during CTIs is shown in Table 6 [1,2]. The radiation doses in our study were smaller than reference 1. The reason being the intermittent CT-fluoroscopy to reduce the scattered radiation exposure from patients used in our study, while in reference 1 it was used continuously [1]. Meanwhile, the physician doses of reference 2 were smaller than this study. By using intermittent CT-fluoroscopy and a low-mA technique, the radiation exposure for physicians can be minimized [2]. However, the numbers of biopsies and occupational doses in our study were larger than the other studies. Moreover, eye and neck proximal doses indicated a much higher level than their distal doses. Thus, we recommend that physicians who participate in CTIs should determine the proximal side of dosimeter placement to evaluate accurate radiation dose [23].

In summary, our study revealed the scattered radiation distribution and occupational doses in MDCT-guided interventions. Our studies lead us to recommend reducing/optimizing the radiation dose in CTIs in the following ways: (1) Optimizing the side of dosimeter placement; (2) minimizing the acquisition and fluoroscopic time, using a lower-dose (mA) mode; (3) Using a set up best suited for lead acrylic shields; and (4) placing a lead drape over the patient when possible.

This study had some limitations. Our results were evaluated at only one institution and using the same CT scanner. In addition, the X-ray condition of CT-fluoroscopy used the usual protocol of this hospital. Therefore, this may not be a common condition worldwide.

## 5. Conclusions

We measured and analyzed the occupational dose associated with CTIs at our hospital using the same CT scanner. Moreover, our study suggests that physician and staff doses were not significant when the procedures were performed with appropriate radiation protection and low-dose techniques.

## Figures and Tables

**Figure 1 diagnostics-11-00646-f001:**
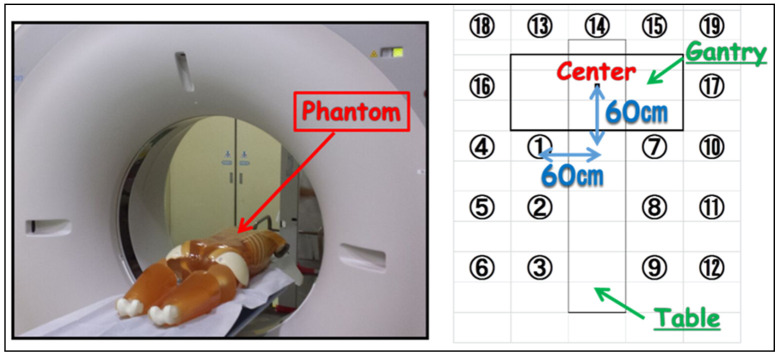
Geometric arrangement of the phantom study (19 points).

**Figure 2 diagnostics-11-00646-f002:**
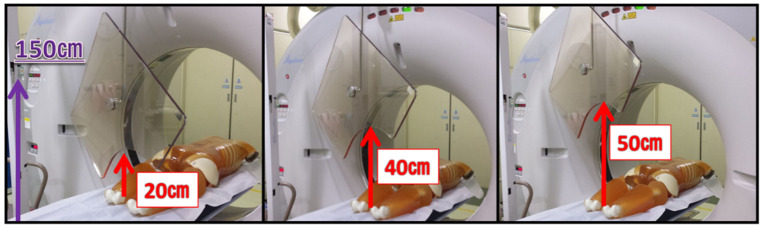
The protective effect of lead acrylic shields at different heights (10, 20, 30, 40, 50, and 70 cm).

**Figure 3 diagnostics-11-00646-f003:**
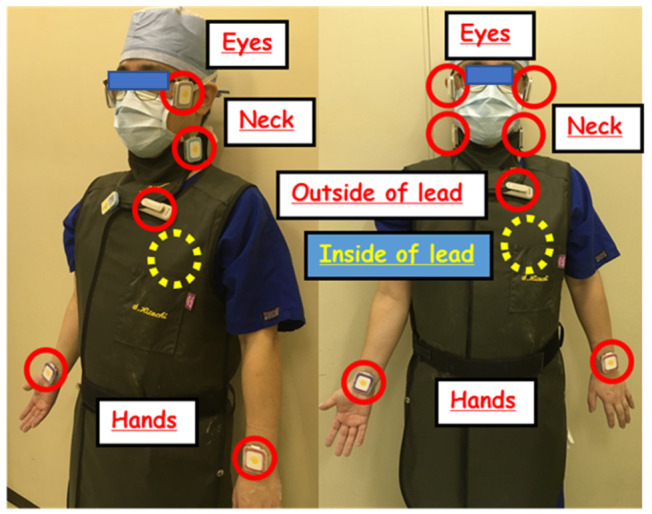
Measurement points during CT-guided intervention (eye, neck, hands, and out/inside of lead apron for estimating the effective dose).

**Figure 4 diagnostics-11-00646-f004:**
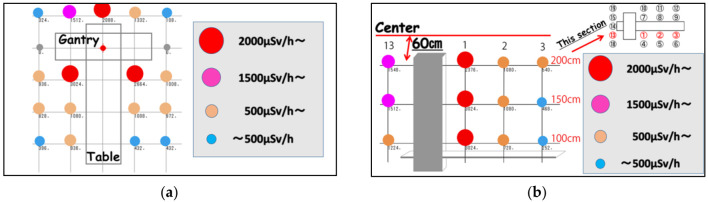
(**a**) The scattered radiation planar distribution at a height of 150 cm; (**b**) The scattered radiation height distribution of points 13, 1, 2 and 3 at a height of 100, 150, and 200 cm.

**Figure 5 diagnostics-11-00646-f005:**
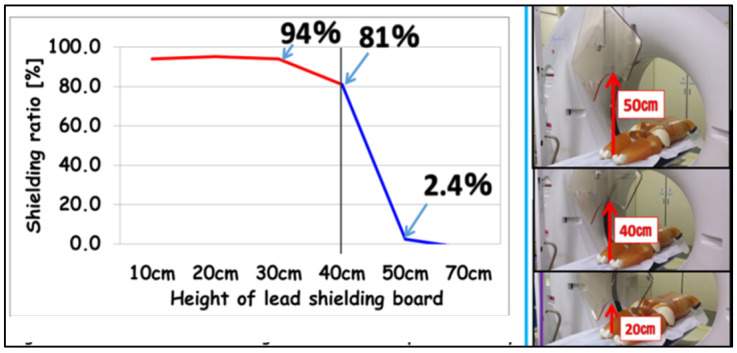
The protective effect of lead acrylic shields regarding height changes.

**Table 1 diagnostics-11-00646-t001:** The details of locations in CT-guided biopsies (232 procedures).

Locations	Bone	Soft Tissue	Sum	Eye Dose [μSv]
Face (Salivary glands et al.)	0	10	10	31.2 ± 32.6
Neck (Cervical spine et al.)	5	7	12	37.3 ± 26.9
Chest (Thoracic spine, Libs et al.)	50	14	64	45.9 ± 43.4
Abdomen (Lumbar spine et al.)	28	11	39	50.8 ± 49.8
Pelvis (Sacrum, Ilium et al.)	36	8	44	32.9 ± 27.3
Upper extremity (Shoulder, Arm et al.)	18	3	21	39.3 ± 28.8
Lower extremity (Thigh, Tibial bone et al.)	39	3	42	32.1 ± 30.2
All procedures	176	56	232	39.1 ± 36.3

**Table 2 diagnostics-11-00646-t002:** Summary of our study in CT-guided biopsies (232 procedures). Ave.: average, SD: standard deviation, CTDI: CT dose index, DLP: dose length product.

Characteristics	Ave. ± SD	Median	Range
Age (years)	52.8 ± 20.9	58.0	1.0~88.0
BMI	23.0 ± 5.2	22.5	15.1~60.2
Acquisitions No.	34.7 ± 23.1	27.0	3.0~160.0
CT-fluoroscopic time (s)	26.6 ± 17.8	20.9	2.3~123.3
CT-fluoroscopic mAs	650.0 ± 598.4	441.0	23.0~4750.0
CTDI vol (mGy)	14.2 ± 39.4	9.0	3.1~606.0
DLP (mGy*cm)	262.4 ± 144.2	217.9	37.3~972.6
Target depth (mm)	66.6 ± 23.6	66.2	11.0~134.3
Physician dose (μSv)			
Eye dose	39.1 ± 36.3	26.7	0.9~285.2
Hand dose	28.6 ± 31.0	19.0	0.0~279.8
Neck dose	23.1 ± 23.7	16.6	0.4~220.7
Effective dose	2.2 ± 2.8	1.3	0.0~16.9
Nurse dose (μSv)			
Neck dose	2.3 ± 5.0	0.7	0.0~55.1
Chest dose	2.4 ± 4.4	1.0	0.0~44.4

**Table 3 diagnostics-11-00646-t003:** Correlation coefficient (r) between physician doses and CT dose-related factors (232 procedures).

Correlation Coefficient (r)	CT-Acquisitions No.	CT-Fluoroscopic Time (s)	CT-Fluoroscopic mAs
Physician dose (μSv)			
Eye dose	0.74	0.73	0.90
Hand dose	0.61	0.60	0.75
Neck dose	0.67	0.67	0.83
Nurse dose (μSv)			
Neck dose	0.29	0.29	0.36
Chest dose	0.36	0.37	0.43

**Table 4 diagnostics-11-00646-t004:** Comparison of the radiation doses and CT-related factors between bone and soft tissue biopsies (Ave. ± SD).

Dosase Groups	Bone	Soft Tissue	Biopsy
Ratio	176	56	*p*-Value
CT-acquisitions No.	36.8 ± 24.9	30.8 ± 19.3	0.074
CT-fluoroscopic time (s)	27.4 ± 18.6	26.3 ± 17.7	0.690
CT-fluoroscopic mAs	698 ± 685	590 ± 440	0.195
CTDI vol (mGy)	10.9 ± 7.6	11.1 ± 6.9	0.864
DLP (mGycm)	263 ± 138	248 ± 129	0.487
Target depth (mm)	66.9 ± 22.6	66.1 ± 26.1	0.836
Physician dose (μSv)			
Eye dose	41.1 ± 37.7	37.1 ± 37.2	0.511
Hand dose	28.6 ± 32.4	28.0 ± 28.8	0.890
Neck dose	24.6 ± 25.6	21.0 ± 21.5	0.323
Effective dose	2.2 ± 2.7	2.5 ± 3.0	0.602
Nurse dose (μSv)			
Neck dose	2.0 ± 3.5	3.4 ± 8.8	0.287
Chest dose	2.1 ± 3.3	3.6 ± 7.5	0.199

**Table 5 diagnostics-11-00646-t005:** Comparison of the radiation doses and CT-related factors between without and with a lead drape in CT-guided biopsies (Ave. ± SD). * *p* < 0.05.

Dosase Groups	Without	With	Lead Drape
Ratio	173	59	*p*-Value
CT-acquisitions No.	35.8 ± 24.2	34.3 ± 22.7	0.674
CT-fluoroscopic time (s)	28.1 ± 19.1	23.7 ± 15.0	0.910
CT-fluoroscopic mAs	688 ± 697	629 ± 439	0.471
CTDI vol (mGy)	10.9 ± 7.6	11.1 ± 6.8	0.854
DLP (mGycm)	259 ± 135	261 ± 137	0.927
Target depth (mm)	66.2 ± 24.2	67.9 ± 21.5	0.627
Physician dose (μSv)			
Eye dose	42.8 ± 40.2	33.1 ± 29.0	0.055
Hand dose	32.7 ± 33.8	17.6 ± 21.4	0.0002 *
Neck dose	25.2 ± 26.0	19.8 ± 20.7	0.124
Effective dose	2.6 ± 3.0	1.5 ± 1.8	0.0008 *
Nurse dose (μSv)			
Neck dose	2.6 ± 6.0	1.8 ± 3.0	0.213
Chest dose	2.7 ± 5.1	1.9 ± 2.9	0.154

**Table 6 diagnostics-11-00646-t006:** Comparison between other studies and our study in CT-guided biopsies (median value).

	Reference [1]	Reference [2]	Our Study
No. of cases	N = 82	N = 85	N = 232
Fluoroscopic time (sec)	-	17.9	20.9
Proximal eye dose (μSv)	210	10	27.1
Distal eye dose (μSv)	-	-	2.4
Proximal neck dose (μSv)	240	-	17.2
Distal neck dose (μSv)	-	-	0.5
Proximal hand dose (μSv)	760	12	18.7
Distal hand dose (μSv)	180	-	11.0
Dosimeter Type	TLD	TLD	RaySafe i2

TLD: Thermoluminescent dosimeter.

## Data Availability

Data sharing not applicable.

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
