# Peer review of "Occupational Radiation Dose to Eye Lenses in CT-Guided Interventions Using MDCT-Fluoroscopy"

_diagnostics, 2021, doi:10.3390/diagnostics11040646_

Round 1
Reviewer 1 Report
This study examines the radiation dose during CT-guided interventions.
Abstract
You definitely need to elaborate in the abstract on how the doses were measured.
L24 Please delete “obviously”. Further, please give numbers for the “obviously” lower dose in nurses.
Introduction
L33 Most radiologists might know about the benefits of CT-guided interventions. Nonetheless, it is presumptuous to think this counts for every reader. Please elaborate on the “clear benefits”. There is more than enough literature to underline this statement.
L43 Please elaborate a little more the problem of deterministic radiation damage, such as the cataract.
Methods
Where is your ethical approval?
L55 What is the minimal sensitivity of the dosimeters used? Were even the smallest radiation doses measured?
L61 Please revise this sentence. Currently you are saying, that the scattered radiation is changing the height. But I think you are referring to the lead acrylic shields. Further, was the dosimeter placed on the backside of the panel?
Results
L99 That is no sentence. Please delete it.
L100 That belongs to the methods, where you have already covered it. Please delete.
L108 Please refrain from statements like “obviously”.
It would be nice to know, if there are dose differences in regard to the interventions and locations. Especially between Chest and e.g. lower extremity interventions I would expect a huge dose gap due to different procedure times.
Discussion
L125 “Physicians should be appropriate the protection to eye lenses […]” What?? Again, this not an English sentence. Did you want to say, that the maximum eye lens dose could be reached in about 90 minutes without appropriate protection? Please thoroughly revise your language!
L126 The next sentence is, just as the one before, unreadable. What did you want to say?
L135 “Approximately” with a small a
L139 and following Again, there is no English to be seen in this section.
L142 This is a result. Why is this in the discussion section??
L143 No tables in the discussion.
Whole page 6 and 7 are results.
Your study has no limitations? This is pretty presumptuous. What about using just a single scanner with mainly one scanner setting? Are your tube currents and voltages universal for all scanners worldwide? Are there other fluoroscopy protocols used at other sites? What about missing prediction models for deterministic radiation damage? What about comparisons to other studies? Are your findings in line with other studies?
Figures
Good
Tables
Good
English should be improved.
References
Have you actually recited yourself 11 times? I have honestly never seen this in my years as a reviewer. You are all the time citing studies in bulk while not elaborating on them in your paper and inbetween you are hiding 11 (!) self-citations? Delete at least 8 of them or explain IN THE TEXT, why the citation is now important at the specific point.
Author Response
Reviewer 1
Comments and Suggestions for Authors
This study examines the radiation dose during CT-guided interventions.
Thank you for your review and comments. We have read your comments very carefully and revised our manuscript accordingly.
Abstract
You definitely need to elaborate in the abstract on how the doses were measured.
Response 1: Thank you for your comments and suggestion. We added the sentence as suggested,
“We measured the occupational radiation dose (physician and nurse) using a personal dosimeter in real-time and gathered CT-related parameters (fluoroscopy time, mAs, CTDI and DLP) performed consecutive 232 procedures in CT-guided biopsy”.
L24 Please delete “obviously”. Further, please give numbers for the “obviously” lower dose in nurses.
Response 2: Thank you for your comments and suggestion. We deleted “obviously” as suggested and revised the sentence as follows,
“Nurse doses (neck and chest) were lower (2.3 ± 5.0 and 2.4 ± 4.4, respectively) than the physician doses”.
Introduction
L33 Most radiologists might know about the benefits of CT-guided interventions. Nonetheless, it is presumptuous to think this counts for every reader. Please elaborate on the “clear benefits”. There is more than enough literature to underline this statement.
Response 3: Thank you for your comments and suggestion. We added the sentence as follows,
“Thus, many studies have been shown that CTIs are a safe and effective guidance tool for percutaneous interventional procedures compared to those of ultrasonograhy (US). CT-fluoroscopy has the potential to improve the efficacy of CTIs, to reduce the time to place a needle into a lesion for the performance of biopsy, and to reduce procedure times.”.
L43 Please elaborate a little more the problem of deterministic radiation damage, such as the cataract.
Response 4: Thank you for your comments and suggestion. We revised the sentence as follows,
“One of the concerns with the use of CT-fluoroscopy is the high radiation exposure. The physician and staff are at high risk of the tissue reaction (such as cataracts) at locations close to the scanning plane.”.
Methods
Where is your ethical approval?
L55 What is the minimal sensitivity of the dosimeters used? Were even the smallest radiation doses measured?
Response 5: Thank you for your comments and suggestion. We added the sentence as follows,
“The sensitivity of i2 dosimeter ranges from 40 μSv/h to 300 mSv/h”.
L61 Please revise this sentence. Currently you are saying, that the scattered radiation is changing the height. But I think you are referring to the lead acrylic shields. Further, was the dosimeter placed on the backside of the panel?
Response 6: Thank you for your comments and suggestion. We revised the sentence as follows,
“In order to determine a protective effect of lead acrylic shields suspended from the ceiling (KYOWAGLAS-XA, lead equivalent: 0.5 mmPb, 65 × 70 cm; Osaka, Japan), we measured scattered radiation at different heights (10, 20, 30, 40, 50, and 70 cm), and i2 dosimeter placed at position â‘ of 150cm height from the floor”.
Results
L99 That is no sentence. Please delete it.
Response 7: Thank you for your comments and suggestion. This sentence is the caption of results 3.2. (Occupational doses (Physician and Nurse) in Clinical Setting)
L100 That belongs to the methods, where you have already covered it. Please delete.
Response 8: Thank you for your comments and suggestion. We deleted as suggested.
L108 Please refrain from statements like “obviously”.
Response 9: Thank you for your comments and suggestion. We deleted “obviously” as suggested and revised the sentence as follows,
“Nurse doses (neck and chest) were lower (2.3 ± 5.0 and 2.4 ± 4.4, respectively) than than physician doses”.
It would be nice to know, if there are dose differences in regard to the interventions and locations. Especially between Chest and e.g. lower extremity interventions I would expect a huge dose gap due to different procedure times.
Response 10: Thank you for your comments and suggestion. We added the eye dose in Table 1 and the sentence as follows,
“Table 1 showed the details of locations and eye dose each location for 232 procedures in CT-guided biopsies.”.
Discussion
L125 “Physicians should be appropriate the protection to eye lenses […]” What?? Again, this not an English sentence. Did you want to say, that the maximum eye lens dose could be reached in about 90 minutes without appropriate protection? Please thoroughly revise your language!
Response 11: Thank you for your comments and suggestion. We revised the sentence as follows,
“The eye lens dose for physicians could be reached the revised eye equivalent dose limit (20 mSv/y) in about 90 minutes if the appropriate protection tools have not been used.”.
L126 The next sentence is, just as the one before, unreadable. What did you want to say?
Response 12: Thank you for your comments and suggestion. We revised the sentence as follows,
“Moreover, since the amount of scattered radiation in the height direction indicated almost the same value, it is attention to radiation exposes for physicians at all height points. Therefore, the physician needs to optimize the eye lens protection such as using the lead acrylic shields or the lead glass eye shields to reduce the lens exposure.”.
L135 “Approximately” with a small a
Response 13: Thank you for your comments and suggestion. We revised as suggested to “approximately ".
L139 and following Again, there is no English to be seen in this section.
Response 14: Thank you for your comments and suggestion. We revised the sentence as follows,
“Since it cannot estimate the nurse dose from CT dose-related factors, the nurse should be appropriate position such as CT gantry sides to reduce exposure of scattered radiation.”.
L142 This is a result. Why is this in the discussion section??
L143 No tables in the discussion.
Whole page 6 and 7 are results.
Response 15: Thank you for your comments and suggestion. We moved this sentence (line 142-151) and Table 4, 5 to result sections as suggested.
Your study has no limitations? This is pretty presumptuous. What about using just a single scanner with mainly one scanner setting? Are your tube currents and voltages universal for all scanners worldwide? Are there other fluoroscopy protocols used at other sites? What about missing prediction models for deterministic radiation damage? What about comparisons to other studies? Are your findings in line with other studies?
Response 16: Thank you for your comments and suggestion. We added the limitations as follows,
“This study has some limitations. Our results were only evaluated at our institution the same CT-scanner. In addition, the X-ray condition of CT-fluoroscopy used usual protocol of my hospital. ”.
References
Have you actually recited yourself 11 times? I have honestly never seen this in my years as a reviewer. You are all the time citing studies in bulk while not elaborating on them in your paper and inbetween you are hiding 11 (!) self-citations? Delete at least 8 of them or explain IN THE TEXT, why the citation is now important at the specific point.
Response 17: Thank you for your comments and suggestion. We deleted seven papers as suggested
Reviewer 2 Report
Congratulations on gathering interesting data on occupational doses in CT-guided interventions. Unfortunately, I cannot recommend to publish the paper in its current form. Methods need better description, results need better discussion.
Methods: Did physicians use protective glasses? If they did, was the protection effect included in calculation of eye dose? (the dosimeter was obviously not placed behind the glasses)
Line 40-41. The reduction in dose limit happened in 2012, it is not a new information anymore.
Line 70-71. Information about CT system and hospital could be placed at the beginning of "Materials and Methods" section (if it applies both to 2.1 and 2.2).
Line 88. "3000 microSv/h" would be for a continouos 1h of radiation, or for the typical hour of work ?
Table 2. How was effective dose calculated from the dosimeter readings? Please describe it in methods.
Table 3. Does red color mean "statistical significant"? Please explain. Have you checked correlation between physician doses and really DOSE-related factors, as CTDI and DLP? If not - why?
Table 4, Table 5, Line 147. What statistical tests were used? Please describe in methods.
Table 6. There is a tenfold difference between results in reference 1 and results in reference 2 and present study. How can it be explained? Different medical procedures? Different methods, equipment..? Please discuss that.
In the "phantom study" authors evaluated the effect of lead acrylic shields placed at different heights, but in the "occupational doses" part they did not collected data on the height. On the other side, in the "occupational doses" they noted use of lead drape over the patient, but the effect of lead drape was not measured in "phantom study" part.
Conclusions: First sentence of conclusions is not really true. Authors have not "clarified the occupational dose associated with CTIs in MDCT-fluoroscopy", they rather measured and analyzed occupational doses for one CT, in one hospital. Second sentence of conclusions is certainly true, but is not a conclusion. Last sentence - how do you judge, if the doses are significant or not?
Whole paper certainly needs a language correction. In fact, some sentences are hard to understand.
Line 61-62. "scattered radiation that changed the height" - hard to understand. Needs to be rewritten in proper English.
Equation on page 2 - "sheilding" in denominator needs correction
Figure 2 caption - "changing the height" could be replaced with "at different heights"
Line 73-74: could be rewritten as "...for consecutive 232 procedures in CT-guided biopsy performed from May 73 2014 to January 2017."
Line 94. "but it was very low, 50 cm height later" - hard to understand, needs correction.
Table 1 caption. Maybe "details" instead of "detailed"?
Table 1. I guess authors meant "lumbar", not "lumber".
Line 123. "It was reported" or "It has been reported".
Line 125-126. Hard to understand.
Line 130. Hard to understand. Nurse does no "move the gantry sides"...
Line 134. Hard to understand. "in order to be close to the source of scattered radiation" - maybe you meant "if the physician is close.."?
Line 140-141. Hard to understand.
Author Response
Reviewer 2
Comments and Suggestions for Authors
Congratulations on gathering interesting data on occupational doses in CT-guided interventions. Unfortunately, I cannot recommend to publish the paper in its current form. Methods need better description, results need better discussion.
Thank you for your review and comments. We have read your comments very carefully and revised our manuscript accordingly.
Methods: Did physicians use protective glasses? If they did, was the protection effect included in calculation of eye dose? (the dosimeter was obviously not placed behind the glasses)
Response 1: Thank you for your comments and suggestion. We added the sentence as follows,
“The physicians did not use protective eyeglasses and lead acrylic shields in this clinical setting.”.
Line 40-41. The reduction in dose limit happened in 2012, it is not a new information anymore.
Response 2: Thank you for your comments and suggestion. We revised the sentence as follows,
“In 2012, the International Commission on Radiological Protection (ICRP) recommended that the occupational dose limit to the eye lens is reduced by approximately 1/8 (from 150mSv/year to 20mSv/year, averaged over defined 5 years, with no exceeding 50mSv in any year) [14]. Moreover, the International Atomic Energy Agency (IAEA) has adopted a new eye dose limit by ICRP [15], and then many countries are considering adoption in regulation. In Japan, since the new eye dose limit would be adopted in April 2021, the evaluation of occupational dose is important.”.
Line 70-71. Information about CT system and hospital could be placed at the beginning of "Materials and Methods" section (if it applies both to 2.1 and 2.2).
Response 3: Thank you for your comments and suggestion. We placed this sentence at the beginning of "Materials and Methods" section as suggested.
Line 88. "3000 microSv/h" would be for a continouos 1h of radiation, or for the typical hour of work ?
Response 4: Thank you for your comments and suggestion. "3000 microSv/h" estimated from the scattered radiation dose of 10sec fluoroscopic time.
Table 2. How was effective dose calculated from the dosimeter readings? Please describe it in methods.
Response 5: Thank you for your comments and suggestion. We added the sentence as follows,
“The effective dose was evaluated by the following equation (2), Ha; outside dose of lead apron, Hb; inside dose of lead apron.
Effective dose =0.11Ha+0.89Hb (2)”.
Table 3. Does red color mean "statistical significant"? Please explain. Have you checked correlation between physician doses and really DOSE-related factors, as CTDI and DLP? If not - why?
Response 6: Thank you for your comments and suggestion. We deleted red color and added the sentence as follows,
“There was no significant correlations between occupational dose and CTDI vol, DLP.”.
Table 4, Table 5, Line 147. What statistical tests were used? Please describe in methods.
Response 7: Thank you for your comments and suggestion. We added the statical analysis section as follows,
“Pearson correlatin coefficients analysis was used to determine whether CT-related parameters were linearly related to occupational doses in Table 3. Weltch’s t-test was used to compare between dosage groups of the bone and soft tissue in Table 4 and to compare between dosage groups of without and with lead drape in Table 5.
All statistical analyses were performed using the JMP Pro 15 software (SAS Institute Inc, NC, USA). The statistical significance was defined as P < 0.05.”.
Table 6. There is a tenfold difference between results in reference 1 and results in reference 2 and present study. How can it be explained? Different medical procedures? Different methods, equipment..? Please discuss that.
Response 8: Thank you for your comments and suggestion. We revised the sentence as follows,
“The radiation doses in our study were smaller than the reference 1. This reason is used intermittent CT-fluoroscopy in our study, although reference 1 is used coutinuous it [1]. Meanwhile, the physician doses of reference 2 were smaller than this study. By using intermittent CT-fluoroscopy and a low-mA technique, the radiation exposure for physician can be minimized [2].”.
In the "phantom study" authors evaluated the effect of lead acrylic shields placed at different heights, but in the "occupational doses" part they did not collected data on the height. On the other side, in the "occupational doses" they noted use of lead drape over the patient, but the effect of lead drape was not measured in "phantom study" part.
Response 9: Thank you for your comments and suggestion. We added the sentence as follows. We did not evaluate the effect of lead drape in phantom study.
“The physicians did not use protective eyeglasses and lead acrylic shields in this clinical setting.”.
Conclusions: First sentence of conclusions is not really true. Authors have not "clarified the occupational dose associated with CTIs in MDCT-fluoroscopy", they rather measured and analyzed occupational doses for one CT, in one hospital. Second sentence of conclusions is certainly true, but is not a conclusion. Last sentence - how do you judge, if the doses are significant or not?
Response 10: Thank you for your comments and suggestion. We revised the sentence of discusstion and conclusions as follows,
Discusstion; “However, when assessed by average eye dose (about 40 μSv/procedure), the new eye dose limit would be reached in approximately 500 procedures/year. Some studies have report-ed that the shielding effect of the radiation protective eyeglasses was approximately 50 ~ 60% [16, 22-23]. In this way, if protective eyeglasses are used during CTIs, up to about 1000 procedures could be performed annually. Thus, the physician doses was not significant when the procedures were performed with appropriate radiation protection..”
Conclusions; “We measured and analyzed the occupational dose associated with CTIs at our hospital the same CT scanner. Moreover, our study suggests that physician and staff doses were not significant when the procedures were performed with appropriate radiation protection and low-dose tech-niques.”
Whole paper certainly needs a language correction. In fact, some sentences are hard to understand.
Line 61-62. "scattered radiation that changed the height" - hard to understand. Needs to be rewritten in proper English.
Response 11: Thank you for your comments and suggestion. We revised the sentence as follows,
“In order to determine a protective effect of lead acrylic shields suspended from the ceiling (KYOWAGLAS-XA, lead equivalent: 0.5 mmPb, 65 × 70 cm; Osaka, Japan), we measured scattered radiation at different heights (10, 20, 30, 40, 50, and 70 cm), and i2 dosimeter placed at â‘ position of 150cm height from the floor”.
Equation on page 2 - "sheilding" in denominator needs correction
Response 12: Thank you for your comments and suggestion. We revised the equation as suggested to "shielding".
Figure 2 caption - "changing the height" could be replaced with "at different heights"
Response 13: Thank you for your comments and suggestion. We revised the sentence as suggested "at different heights".
Line 73-74: could be rewritten as "...for consecutive 232 procedures in CT-guided biopsy performed from May 73 2014 to January 2017."
Response 14: Thank you for your comments and suggestion. We revised the sentence as follows,
“It evaluated the occupational radiation doses (physician and nurse) and CT-related parameters (fluoroscopic time, fluoroscopic mAs, CTDI and DLP et al) for consecutive 232 procedures in CT-guided biopsy performed from May 2014 to January 2017.”.
Line 94. "but it was very low, 50 cm height later" - hard to understand, needs correction.
Response 15: Thank you for your comments and suggestion. We revised the sentence as follows,
“The shielding ratio regarding height changes was approximately 80% until 40cm height, but 50 cm height later was very low (Figure 5)”.
Table 1 caption. Maybe "details" instead of "detailed"?
Response 16: Thank you for your comments and suggestion. We revised as suggested to "details".
Table 1. I guess authors meant "lumbar", not "lumber".
Response 17: Thank you for your comments and suggestion. We revised as suggested to " lumbar ".
Line 123. "It was reported" or "It has been reported".
Response 18: Thank you for your comments and suggestion. We revised as suggested to " It has been reported ".
Line 125-126. Hard to understand.
Response 19: Thank you for your comments and suggestion. We revised the sentence as follows,
“The eye lens dose for physicians could be reached the revised eye equivalent dose limit (20 mSv/y) in about 90 minutes if the appropriate protection tools have not been used.”.
Line 130. Hard to understand. Nurse does no "move the gantry sides"...
Response 20: Thank you for your comments and suggestion. We revised the sentence as follows,
“We also recommend that medical staff such as nurse move to the gantry sides (0mSv/h) as nearly as possible for reducing exposure of radiation.”.
Line 134. Hard to understand. "in order to be close to the source of scattered radiation" - maybe you meant "if the physician is close.."?
Response 21: Thank you for your comments and suggestion. We revised the sentence as follows,
“Our clinical results, we revealed that the maximum eye lens dose for physician was about 300 μSv/procedure if the physician is close to the source of scattered radiation without radiation protective tools.”.
Line 140-141. Hard to understand.
Response 22: Thank you for your comments and suggestion. We revised the sentence as follow,
“Since it cannot estimate the nurse dose from CT dose-related factors, the nurse should be appropriate position such as CT gantry sides to reduce exposure of scattered radiation.”.

Round 2
Reviewer 2 Report
Congratulations on your work!
Author Response
thank you for your comments